# A 10 km daily-level ultraviolet radiation predicting dataset based on machine learning models in China from 2005 to 2020

Yichen Jiang[1], Su Shi[1], Xinyue Li[1], Chang Xu[1], Haidong Kan[1, 2], Bo Hu[3, *], Xia Meng[1, 2, **]

[1]School of Public Health, Key Laboratory of Public Health Safety of the Ministry of Education and Key Laboratory of Health Technology Assessment of the Ministry of Health, Fudan University, Shanghai, 200032, China

[2]Shanghai Key Laboratory of Meteorology and Health IRDR International Center of Excellence on Risk Interconnectivity and Governance on Weather/Climate Extremes Impact and Public Health WMO/IGAC MAP-AQ Asian Office Shanghai, Fudan University, Shanghai, China

[3]State Key Laboratory of Atmospheric Boundary Layer Physics and Atmospheric Chemistry, Institute of Atmospheric Physics, Chinese Academy of Sciences, Beijing, People's Republic of China

*Correspondence to*: Xia Meng (mengxia@fudan.edu.cn); Bo Hu (hub@post.iap.ac.cn)

**Abstract.** Ultraviolet (UV) radiation is closely related to health; however, limited measurements have hindered further investigation of its health effects in China. Machine learning algorithms have been widely used to predict environmental factors with high accuracy, but limited studies have implemented it for UV radiation. The main aim of this study is to develop UV radiation prediction model using the random forest approach and predict the UV radiation at daily and 10 km resolution in mainland China from 2005 to 2020. The model was developed with multiple predictors such as UV radiation data from satellites as independent variables and ground UV radiation measurements from monitoring stations as the dependent variable. Missing satellite-based UV radiation data were obtained using the three-day moving average method. The model performance was evaluated using multiple cross-validation (CV) methods. The overall $R^2$ and root mean square error between measured and predicted UV radiation from model development and model 10-fold CV were 0.97 and 15.64 W m$^{-2}$ and 0.83 and 37.44 W m$^{-2}$ at daily level, respectively. The model that incorporated erythemal daily dose (EDD) retrieved from the Ozone Monitoring Instrument (OMI) had a higher prediction accuracy than that without it. Based on predictions of UV radiation at a daily level, 10 km spatial resolution, and nearly 100% spatiotemporal coverage, we found that UV radiation increased by 4.20%, PM$_{2.5}$ levels decreased by 48.51%, and O$_3$ levels increased by 22.70%, respectively, from 2013–2020, suggesting a potential correlation among these environmental factors. The uneven spatial distribution of UV radiation was associated with factors such as latitude, elevation, meteorological factors, and season. The eastern areas of China pose a higher risk due to both high population density and high UV radiation intensity. Using machine learning algorithm, this study generated gridded UV radiation dataset with extensive spatiotemporal coverage, which can be utilized for future health-related research. This dataset is freely available at https://doi.org/10.5281/zenodo.10884591 (Jiang et al., 2024).

## 1 Introduction

Ultraviolet (UV) radiation is a crucial environmental factor closely associated with human health (Brenner and Hearing, 2008;

Narayanan et al., 2010). Previous studies have confirmed the hazardous effects of UV radiation on skin cancer (Griffin et al., 2023; Vienneau et al., 2017) but inconsistent results have been reported regarding the adverse effects of UV radiation on eye diseases (Lagreze et al., 2017; Tian et al., 2018; Wolffsohn et al., 2022) and the health benefits of moderate UV radiation (Boscoe and Schymura, 2006; Vopham et al., 2017; Swaminathan et al., 2019). Further studies are required to ascertain the effects of UV radiation on human health; however, the lack of high-accurate exposure data of UV radiation hinders such health-related investigations.

Exposure assessment methods used in previous health studies on UV radiation mainly include the following: first, the UV index, a frequently used proxy for UV radiation in epidemiological studies (Thayer, 2014; Marson et al., 2021; Walls et al., 2013). It predicts UV radiation levels on a scale of 1 to 11+. Although the UV index is easy to interpret, converting continuous measurements of UV radiation to the UV index results in the loss of numerical information. Second, satellite remote sensing data, often used to estimate UV radiation exposure. For example, erythemal UV irradiance from the Total Ozone Mapping Spectrometer (TOMS), despite being one of the initial instruments for evaluating the UV radiation backscattered by the Earth's atmospheric layers, it exhibits lower spatial resolution of 50 km×50 km, and it has limited accuracy. (Boscoe and Schymura, 2006; Mohr et al., 2008; Lin et al., 2012; Zhou et al., 2019). Erythemal daily dose (EDD) retrieved from the Ozone Monitoring Instrument (OMI) can be utilized to evaluate the UV radiation exposure level with higher spatiotemporal resolution and was employed in the United States to represent ground UV radiation levels and identify hotspots for skin cancer (Zhou et al., 2019; Deng et al., 2021). However, missing values of the OMI EDD data were non-random. Especially since 2008, the field of view of the instrument has been partially obstructed by the peeling of the spacecraft's protective film, leading to data loss in the center-right section of each observational swath. This has greatly increased the missing rate of OMI EDD data, posing a challenge to the accuracy of exposure assessments in epidemiological studies (Mcpeters et al., 2015). Third, personal dosimeters, often worn to measure individual exposure (Stump et al., 2023; Grandahl et al., 2018). Although the data quality from this method is high, the costs are substantial, making it difficult to apply in large-population studies. Therefore, UV radiation data with higher accuracy and spatiotemporal resolution are required to support further exposure assessments.

The enrichment of data resources and improvements in computing power have led to the development of machine learning algorithms. Machine learning algorithms can integrate data from multiple sources to predict environmental factors with high quality (Chen et al., 2021; Zhu et al., 2022; Liu et al., 2022). However, empirical or statistical models are generally used for UV radiation prediction (González-Rodríguez et al., 2022; Vopham et al., 2016; Pei and He, 2019; Liu et al., 2017). In recent years, some pioneering studies have employed machine learning algorithms to predict UV radiation in China (Wu et al., 2022; Qin et al., 2020). The spatiotemporal resolution of the predictions of one study was relatively low (0.50° × 0.625°) (Qin et al., 2020), while another produced UV radiation predictions with significant missingness due to missing data of one predictor (aerosol optical depth from satellite), which may lead to seasonal bias in the UV radiation assessment (Wu et al., 2022). In

addition, these studies did not include direct measurements of UV radiation from satellites, such as the OMI EDD, which has been proven to be an effective predictor of UV radiation evaluation (Zhou et al., 2019; Deng et al., 2021). Satellite-based measurements can be used as one of the "real" measurements of UV radiation, which can help constrain the overfitting of the model in spatiotemporal extrapolation. Overall, further studies are required to add more evidence to the model development of UV radiation using advanced algorithms and comprehensive predictors.

Therefore, this study aimed to develop a random forest model, one of the machine learning algorithms to predict UV radiation in mainland China at a daily level and a spatial resolution of 10 km in 2005–2020. Multiple predictors, including satellite-based UV radiation, UV radiation simulations, parameters from reanalysis meteorological datasets, and other variables, were included in the model development. The missing satellite-based UV radiation were filled to improve the spatial coverage of the final UV radiation predictions. Finally, based on predictions with relatively high spatiotemporal resolution and a long period, temporal and spatial trends as well as hotspots of UV radiation were identified in mainland China.

## 2 Data and methods

### 2.1 Data

#### 2.1.1 Ground UV radiation measurements

The Chinese Ecosystem Research Network (CERN) has been observing UV radiation since 2004 (Liu et al., 2017). The monitoring data are available online at http://www.cern.ac.cn/. Hourly monitoring data on UV radiation from 40 ground-based stations between 2005 and 2015, and 36 ground-based stations between 2016 and 2020, were collected from CERN (Fig. 1). These stations cover eight ecological land cover types across China: urban, agricultural, grassland, forest, lakes, bays, wetlands, and deserts. Daily UV radiation values were calculated by adding the 24 hourly UV radiation values per day. Days with continuous 2 hourly missing or unavailable UV radiation values were excluded.

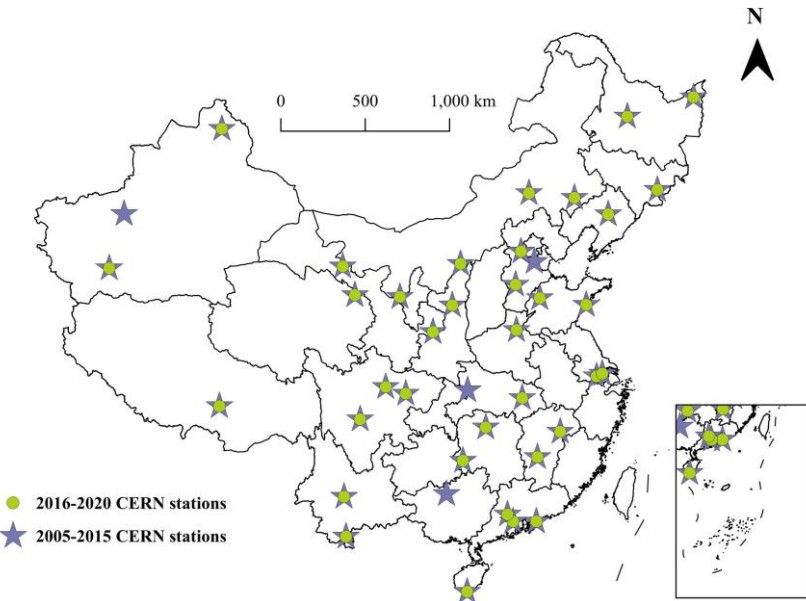

Figure 1. Spatial distributions of CERN stations monitoring UV radiation in China in 2005–2020.

**2.1.2 Predictors directly related to UV radiation**

In this study, Level-2 OMI EDD (v.003) data were utilized as the main predictor of UV radiation, which has a temporal resolution of the daily level and a spatial resolution of 0.25° × 0.25° (Zhou et al., 2019). The OMI EDD represents the overall amount of UV radiation that can cause sunburn during the day. The other predictor was the downward UV radiation at the surface from the fifth-generation European Center for Medium-Range Weather Forecasts Reanalysis at single levels (ERA-5 UV), with a temporal resolution of hourly and a spatial resolution of 0.25° × 0.25° (https://cds.climate.copernicus.eu/). Daily ERA-5 UV data were obtained by adding data over 24 hours for each day. OMI EDD and ERA-5 UV with a spatial resolution of 0.25° × 0.25° were interpolated to 10 km grid cells using the inverse distance weighted (IDW) method.

**2.1.3 Meteorological parameters**

Meteorological parameters that may affect UV radiation were extracted from multiple ERA-5 products (https://cds.climate.copernicus.eu/), according to previous studies (Dieste-Velasco et al., 2023; Hu et al., 2010). The total cloud cover, total column water vapour, and forecast albedo were extracted from a single-level ERA-5 product with a temporal resolution of hourly level and a spatial resolution of 0.25° × 0.25°, and the relative humidity was extracted from the pressure-level ERA-5 product at 1000 hPa with a temporal resolution of hourly level and a spatial resolution of 0.25° × 0.25°. The total precipitation and temperature at 2m were extracted from the ERA5-Land product with a temporal resolution of hourly and a spatial resolution of 0.1° × 0.1°. Regarding temporal resolution, hourly data were converted to daily mean data by averaging the 24-hour data for each day. Concerning spatial resolution, the IDW method was used to interpolate the meteorological parameters to 10 km grid cells.

### 2.1.4 Other predictor variables

Other predictor variables that were incorporated included elevation, solar zenith angle (SZA), ground ozone ($O_3$) concentration, and aerosol optical depth (AOD), which can affect UV radiation levels, according to previous studies (Santos et al., 2011; Habte et al., 2019). Elevation data were derived from the Advanced Spaceborne Thermal Emission and Radiometer (ASTER) Global Digital Elevation Map (GDEM) with a spatial resolution of 30 m (https://asterweb. jpl. nasa. gov/GDEM. asp/). The SZA data were obtained from Aqua (MYD06_L2) with a temporal resolution of daily levels and a spatial resolution of 5 km (https://search.earthdata.nasa.gov). The $O_3$ data were maximum daily 8 h average (MDA8) $O_3$ concentrations predicted based on a random forest model at a daily level and a spatial resolution of $1 \times 1$ km in China (Meng et al., 2022). This study used gridded $O_3$ data instead of $O_3$ monitoring data from station sites, primarily due to considerations of data coverage in both temporal and spatial dimensions. Regarding the temporal coverage, the air quality monitoring network in China has not established until 2013, which could not fully cover the study period of 2005-2020 in this study. For the spatial coverage, the density of air quality monitoring stations is relatively low, with the majority of them are located in urban areas and eastern China, which could not capture the spatial variability within city and reflect the $O_3$ pollution level in rural areas and western regions (Geyh et al., 2000). While the gridded $O_3$ predictions used in this study are available from 2005-2020, have full spatial coverage in mainland China and achieved relatively high accuracy comparing with ground measurements with cross-validation (CV) $R^2$ and root mean square error of 0.80 and 20.93 ug/m$^3$, respectively (Meng et al., 2022). This study also included AOD data from the Multi-Angle Implementation of Atmospheric Correction (MAIAC AOD) algorithm, based on the Moderate Resolution Imaging Spectroradiometer (MODIS), with a temporal resolution of daily levels and a spatial resolution of 1 km (Shi et al., 2022; Meng et al., 2021). The MAIAC AOD values for cloud contamination or land covered by snow were cleaned based on quality assurance (QA) flags. Elevation and SZA were spatially joined and averaged into 10 km grid cells. $O_3$ and MAIAC AOD were obtained by matching 1 km grid cells with 10 km grid cells and then calculating the mean value of the data within the 10 km grid cells.

### 2.1.5 Air pollution data

For comparing the long-term trends of UV radiation and air pollution, fine particulate matter ($PM_{2.5}$) and $O_3$ data were included. $PM_{2.5}$ data were predicted using a random forest model at a daily level and a spatial resolution of $1 \times 1$ km in China (Meng et al., 2021; Shi et al., 2023a; Shi et al., 2023b). The source and spatiotemporal resolution of the $O_3$ data were the same as those in Section 2.1.4 Other predictor variables.

### 2.2 Methods

### 2.2.1 Model development

In recent years, machine learning algorithms have been widely used to predict environmental factors because of their flexibility and excellent data processing capabilities (Corrêa, 2023; Wu et al., 2022). This study utilized a random forest, one of machine learning algorithms, to develop a model for predicting UV radiation in China from 2005–2020. The dependent variable was the daily ground-measured UV radiation, while the independent variables included OMI EDD, ERA-5 UV, elevation, SZA, $O_3$, MAIAC AOD, and meteorological parameters including total cloud cover, relative humidity, total column water vapour, forecast albedo, total precipitation, and temperature at 2 m. Random forest improves the overall prediction performance by building multiple decision trees and combining their results (Breiman, 2001). It uses bootstrap sampling, which draws different subsamples from the original dataset with replacements as training data for each decision tree. During the training process, each decision tree makes predictions for the input data and the final result of the random forest is obtained by averaging the predictions from all trees. Model development was implemented using the "Rborist" package in R version 3.6.3.

OMI EDD is a measurement of UV radiation from a satellite but has non-random missing values due to cloud cover and a technological issue of OMI since 2008, with an averaged missing rate of 23.04% (3.03–35.29%) during all years over the study period (Table A2). We employed the three-day moving average method to fill the OMI EDD values on grid-days with missing data by calculating the mean of the OMI EDD values from the two preceding days if they were available for those grid cells. In the case of grid cells with missing data on consecutive days (more than 1 day), the missing OMI EDD data were not filled in this study. With this method, the missing rate of OMI EDD significantly decreased from 23.04% to 0.62% on average in 2005-2020 (Table A2). 10-fold CV was employed to assess the accuracy of the three-day moving average method for filling the gap of OMI EDD data. In each iteration, 10% of the original OMI EDD data in the dataset were randomly dropped, and the three-day moving average method was applied to fill the missing values. This process was repeated ten times, and the gap-filled OMI EDD values were compared to the corresponding original OMI EDD values. The results of the 10-fold CV are presented in Table A2 in Appendix, with $R^2$ ranging from 0.85 to 0.90 in 2005-2020, indicating the relatively high accuracy of the gap-filling method.

### 2.2.2 Model validation

CV is commonly utilized to assess model performance in regard of overfitting and predicting accuracy, especially in studies of model development for UV radiation (Wu et al., 2022), particulate matter (Chen et al., 2018; Park et al., 2022; Wongnakae et al., 2023), $O_3$ (Hsu et al., 2019; Wu et al., 2021), and nitrogen dioxide (Lu et al., 2021a). In this study, model performance was tested through overall 10-fold CV, temporal 10-fold CV, spatial 10-fold CV, and by-year temporal CV, which is a stricter temporal CV. Overall 10-fold CV is the most commonly used form of CV, offering a dependable evaluation of overall model performance and assessing model overfitting (Wu et al., 2022; Wongnakae et al., 2023; Hsu et al., 2019). Temporal 10-fold CV can evaluate the models' capacity of temporal extrapolation for predicting UV radiation levels on days without measurements (He et al., 2023b; Lu et al., 2021b; Bi et al., 2020; Zhu et al., 2022). Spatial 10-fold CV is able to evaluate the

models' capacity of spatial extrapolation in locations without monitoring stations (Wang et al., 2018; Zhu et al., 2022; Bi et al., 2020). By-year temporal CV can be used to evaluate the predicting accuracy of our models in years out of the study period of model development (Meng et al., 2021; He et al., 2023a; He et al., 2021).

The overall 10-fold CV was conducted by randomly dividing the dataset into ten parts, with nine parts used as a training dataset to train a random forest model and one part used as a test dataset for predictions. This process was repeated ten times and all measurements were compared with the corresponding predictions. Temporal 10-fold CV was done by randomly dividing the dataset into ten parts based on days, in which data on 90% of the days were used to develop a training model to predict UV radiation on the remaining 10% days each time, and this process was repeated ten times. Similarly, spatial 10-fold CV involved randomly dividing the dataset into ten parts based on the locations of monitoring stations, with data from 90% of the sites were used to develop a training model to predict the UV radiation for the remaining 10% of the sites each time and this process was repeated ten times. In order to further validate the predicting accuracy of our models beyond 2005-2020, this study performed another stricter temporal CV, by-year temporal CV, which left an entire year of data as the testing dataset each time, while data from the remaining years are used as the training dataset. Regression $R^2$ and root mean square error (RMSE; the square root of the average of the squared differences between the predictions and measurements) between the UV radiation measurements and predictions from model development and CVs were calculated to indicate the model performance.

### 2.2.3 Impacts of predictors on UV predictions

Two methods were applied to evaluate the impacts of all predictors on UV radiation levels. First, random forest model itself could produce importance rankings of all predictors to evaluate the contribution of each predictor to UV radiation predictions, and this is also one of the advantages of the random forest model. The importance of a predictor was measured by randomly permuting its values and comparing the decrease in predicting accuracy between the predictions before and after the permutation. Second, SHapley Additive exPlanations (SHAP) method can be used to evaluate both contributions and directions of predictors on final predictions (Lundberg and Lee, 2017). SHAP method employs the classic game theory concept of Shapley values to compute the feature importance for a specific machine learning model (Strumbelj and Kononenko, 2010). Aggregating the SHAP values across multiple data points provides a global explanation of the model. In this study, we utilized the SHAP library in Python to interpret impacts of predictors on UV radiation predictions based on a random forest model (Lundberg et al., 2020).

### 3 Results

### 3.1 Description of UV radiation measurements

Table A1 summarizes the statistical descriptions of the average daily mean for UV radiation measurements from CERN from

2005 to 2020. The mean annual value of UV radiation at the monitoring stations was 168.40 W m$^{-2}$, with a standard deviation of 91.39 W m$^{-2}$. During the 16-year period, the minimum level of 155.46 W m$^{-2}$ was recorded in 2010, while the maximum UV radiation level of 190.10 W m$^{-2}$ was recorded in 2020, an increase of 22.28% compared with 2010. UV radiation levels fluctuated between 2005 and 2012; however, the overall trend was relatively stable. From 2013 to 2020, there was a clear increasing trend in UV radiation, which increased by 18.66% during this period.

## 3.2 Model performance

This study compared the levels of UV radiation indicators and measurements of UV radiation. The results indicated an $R^2$ of 0.65 between the ERA-5 UV and UV radiation measurements, and an $R^2$ of 0.55 between the OMI EDD and UV radiation measurements in 2005–2020, indicating that both simulated and satellite remotely sensed UV radiation data could moderately represent ground UV radiation levels.

The overall $R^2$ and RMSE of model development between measured and predicted UV radiation were 0.97 and 15.64 W m$^{-2}$ at the daily level, respectively. Fig. 2 shows the scatter density plots between the measurements and CV predictions of UV radiation at the daily level, including the overall CV (a), spatial CV (b), temporal CV (c), and by-year temporal CV (d). From the density scatter plots, it can be seen that most of the measured-predicted pairs from CV fell on the 1:1 line, indicating relatively high consistency between the measurements and CV predictions. The CV $R^2$ (RMSE) values between measured and predicted UV radiation were 0.83 (37.44 W m$^{-2}$) for overall CV, 0.75 (45.56 W m$^{-2}$) for spatial CV, 0.83 (37.48 W m$^{-2}$) for temporal CV, and 0.82 (38.86 W m$^{-2}$) for by-year CV at daily level, and 0.91 (21.01 W m$^{-2}$), 0.81 (31.14 W m$^{-2}$), 0.91 (21.05 W m$^{-2}$), 0.89 (22.90 W m$^{-2}$) at monthly level for overall, spatial, temporal and by-year temporal CV, respectively. Fig. 3 shows the temporal trend of monthly average values for predicted and measured UV radiation at monitoring stations from 2005 to 2020, which also indicates high consistency, although the predictions tended to overestimate UV radiation when it was low and underestimate UV radiation when it was high.

Fig. A1 illustrates that, with other predictors held constant, the inclusion of OMI EDD as a predictor in the model yielded an overall CV $R^2$ (RMSE) of 0.83 (37.44 W m$^{-2}$), compared to 0.81 (39.18 W m$^{-2}$) when OMI EDD was not included.

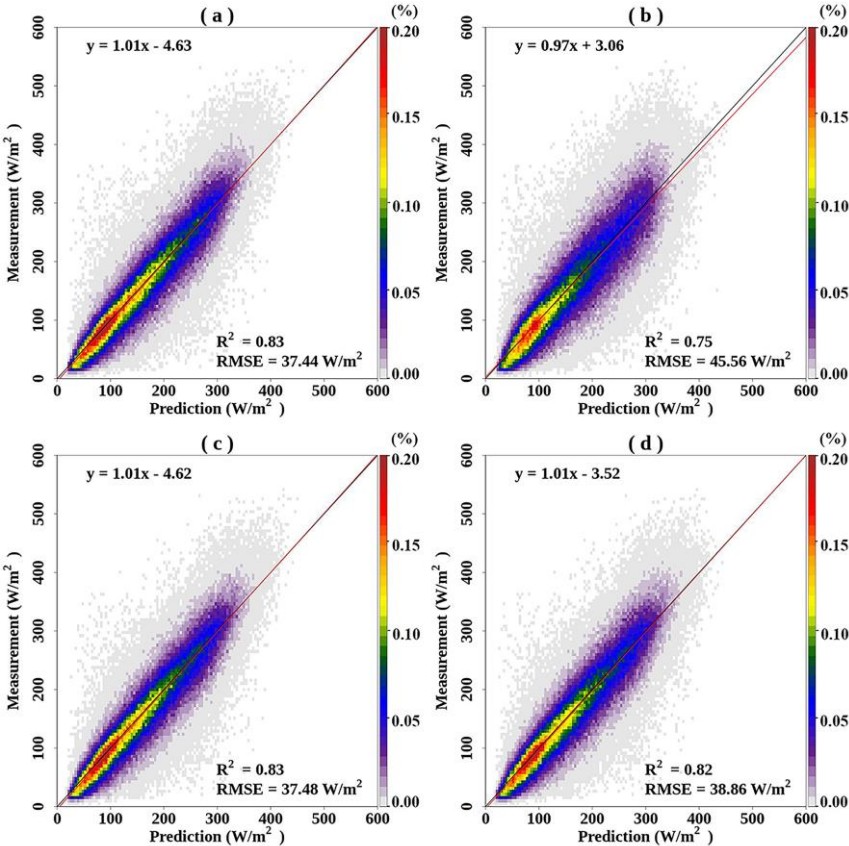

Figure 2. Density scatter plots and linear regressions between measurements and predictions of UV radiation at a daily level based on a random forest model during 2005–2020: Overall CV(a), spatial CV(b), temporal CV(c), and by year temporal CV(d).

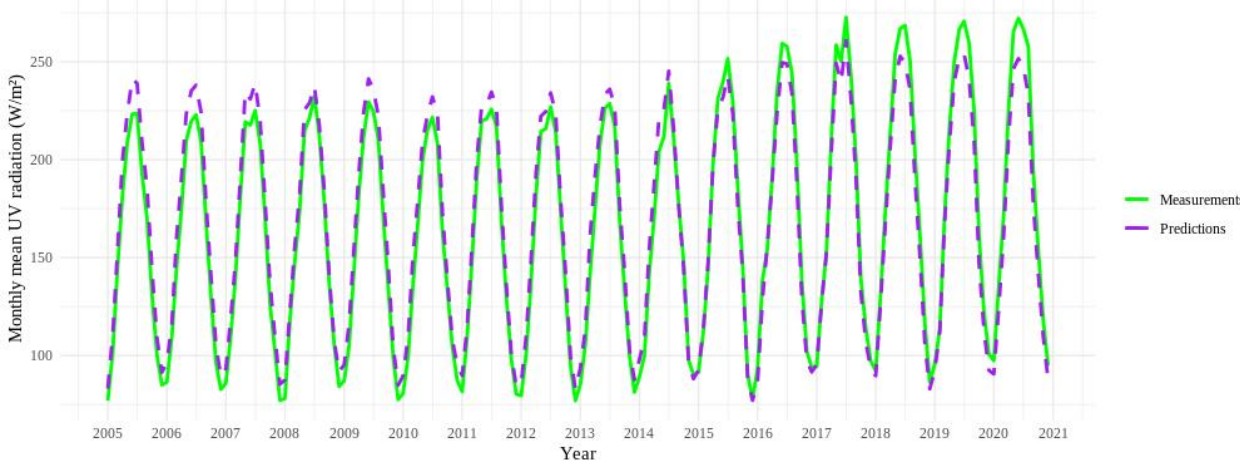

Figure 3. Time series plot of monthly mean UV radiation for measurements (green line) and predictions (purple dash) at monitoring stations during 2005–2020.

### 3.3. Impacts of predictors on UV radiation predictions

Fig. A2 shows the importance ranking of all predictors produced by the random forest model itself that ERA-5 UV, OMI EDD, and MAIAC AOD were the most important predictors of UV radiation. Fig. 4 shows the SHAP summary plot and feature

importance, which were the same with that from the random forest method. SHAP method also provided evaluation on the

impact directions of predictors on UV radiation predictions. In Fig. 4a, each point represents a sample from the dataset. The color of each point indicates the magnitude of the predictor, with redder values indicating higher values and bluer indicating lower values. For example, ERA-5 UV and OMI EDD exerted the most substantial impact and similar impact directions on UV radiation predictions. High values of ERA-5 UV and OMI EDD increased the predicted UV radiation predictions, whereas low values decreased UV radiation predictions. Ambient aerosols (MAIAC AOD) and $O_3$ levels showed opposite effects on UV radiation predictions based on SHAP method. Higher MAIAC AOD values displayed higher negative SHAP values, meaning that higher MAIAC AOD values tended to associate with decreased UV radiation levels. Conversely, High $O_3$ levels corresponded to positive SHAP values, indicating that high $O_3$ levels were associated with high UV radiation predictions.

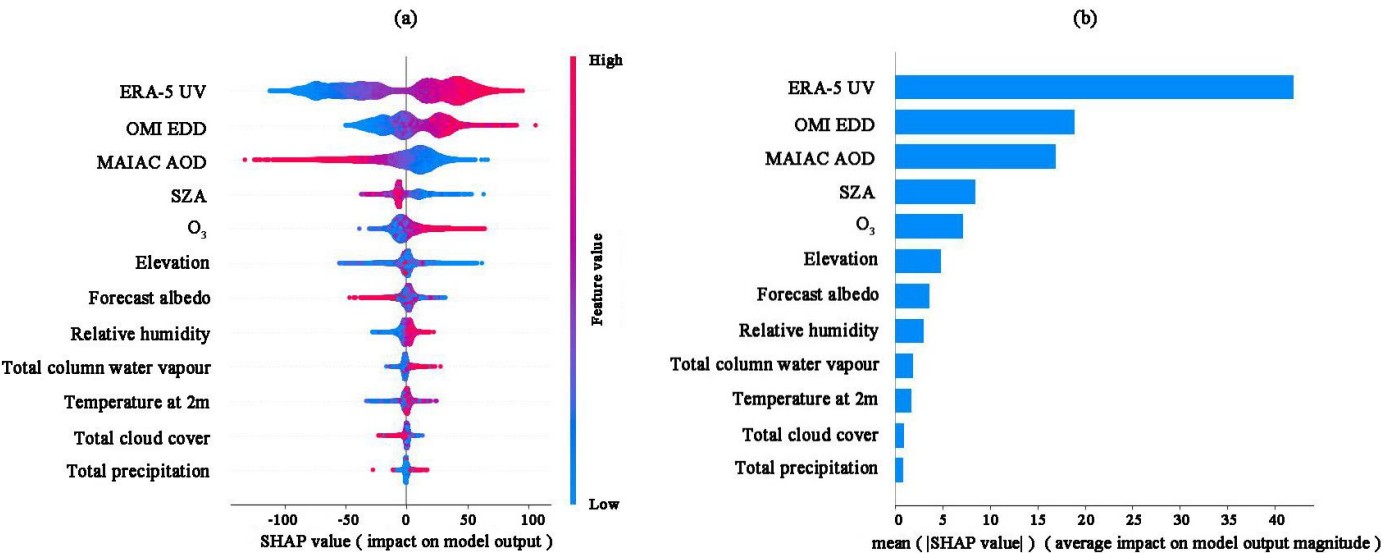

Figure 4. Impacts of predictors on UV radiation predictions based on SHAP method (a); importance ranking of predictors for predicting UV radiation levels, calculated by taking the average of the absolute SHAP values (b).

### 3.4 Spatiotemporal distributions of UV radiation based on predictions

The spatial distribution of annual average UV radiation based on predictions from 2005 to 2020 is shown in Fig. A3 for each year and in Fig. 5 for the average values from 2005 to 2020, indicating an uneven spatial distribution of UV radiation in China associated with factors such as latitude and elevation (Fig. A4) and meteorological factors. On one hand, UV radiation was stronger in the southern region at lower latitudes than in the northern region at higher latitudes. For example, in subregion G in Fig. 5, located at the southernmost latitude in mainland China (~18° N), the UV radiation value was 205.86 W m$^{-2}$; 1.46 times that in subregion A, situated at the northernmost latitude in China (~50° N). On the other hand, UV radiation was higher in western regions, with higher elevation, than in regions with lower elevation, for example, subregion C, with an average elevation of 4730 m, had the highest UV radiation level of 228.36 W m$^{-2}$; 1.50 times that of subregion E, with an average elevation of 5 m. However, because of the influence of climatic factors, the relationship between UV radiation and latitude as

well as elevation may vary in some regions. For example, subregions D and F have similar elevations and latitudes but UV radiation at subregion F was 152.14 W m$^{-2}$; 14.29% higher than that at D. Fig. A5 shows the population density, indicating that although subregion C had the highest UV radiation in China, its population is sparse, while the southeastern coastal areas of China, with dense populations, had relatively strong UV radiation and thus a relatively higher population exposure risk.

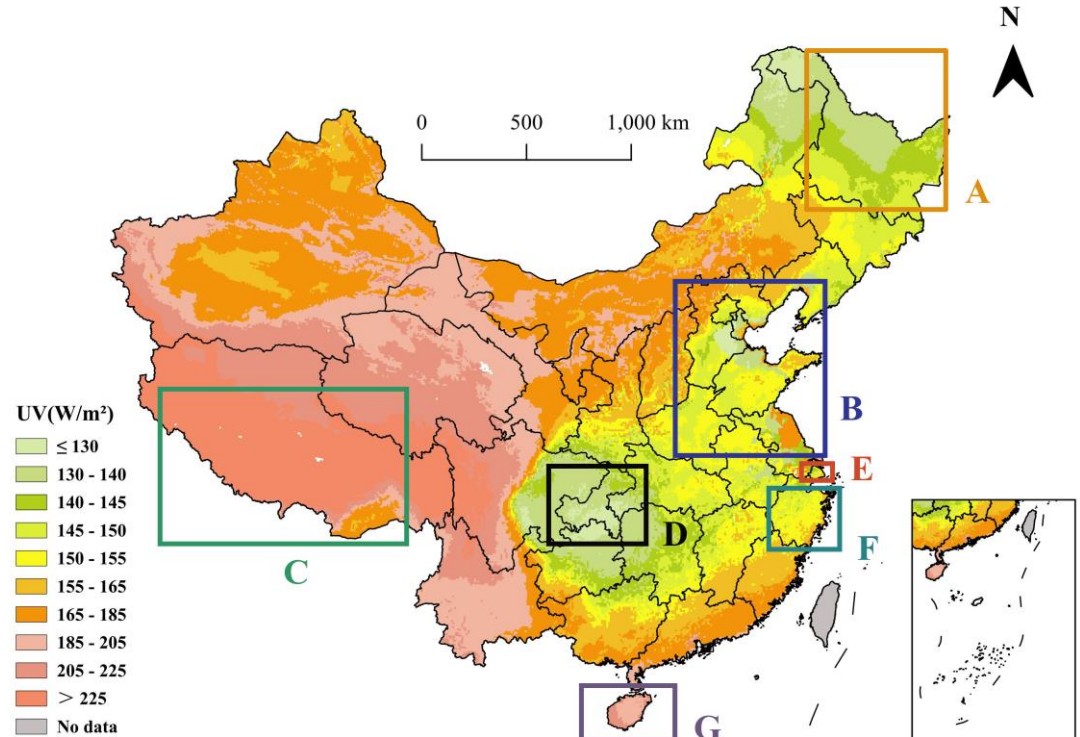

Figure 5. Spatial distribution of averaged annual-mean UV radiation during 2005–2020. Heilongjiang Province (A), North China Plain (B), Tibet Autonomous Region (C), Chongqing City (D), Shanghai City (E), Zhejiang Province (F), and Hainan Province (G).

The inter-annual and intra-annual trends in UV radiation are shown in Fig. 6. For long-term temporal trends, UV radiation experienced slight fluctuations from 2005 to 2014 but remained relatively stable and then increased from 2015. Fig. 6a depicts the trends in the changes in UV radiation, O$_3$, and PM$_{2.5}$ across mainland China from 2013 to 2020, showing that PM$_{2.5}$ demonstrated a prominent downward trend, whereas both UV radiation and O$_3$ exhibited noticeable upward trends during this period. In comparison to 2013, UV radiation increased by 4.20% nationwide in 2020, rising from 176.68 W m$^{-2}$ to 184.10 W m$^{-2}$, O$_3$ increased by 22.70%, while PM$_{2.5}$ decreased by 48.51%. Additionally, Fig. A3 shows that the North China Plain (subregion B in Fig. 5) increased the most significant, with UV radiation increasing by 7.13% from 2013 to 2020, which was 1.70 times the national growth rate. Regarding intra-annual variation, UV radiation exhibited a clear seasonal trend, with significantly higher levels during summer than during winter. It was highest in July, with an average value of 253.02 W m$^{-2}$ in 2005–2020, and then gradually decreased, reaching its lowest value in December, with an average of 89.81 W m$^{-2}$. Additionally, Fig. 6(c)–(f) illustrate the varying spatial trends of UV radiation across different seasons. In spring, the intensity of UV radiation in the northern regions surpassed that in most of the southern areas. During summer, the

UV radiation across mainland China consistently exceeds 162 W m$^{-2}$. The spatial distribution of the UV radiation intensity was primarily affected by elevation and latitude in autumn. In winter, except for in some areas in western China, the UV radiation levels remained below 140 W m$^{-2}$.

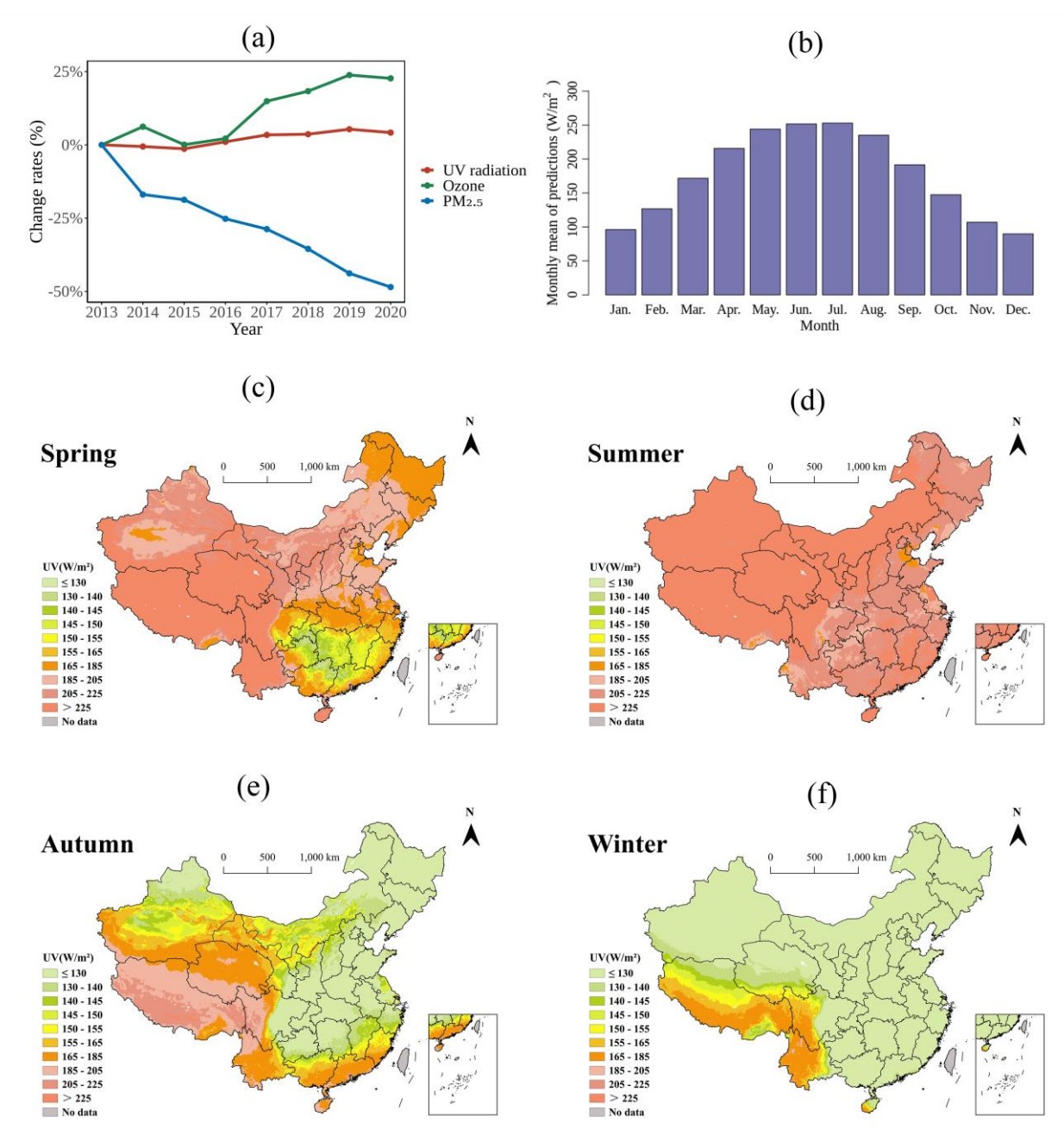

Figure 6. Inter-annual and intra-annual variation of UV radiation based on predictions in mainland China. Annual change rates of UV radiation, O$_3$, PM$_{2.5}$ in mainland China from 2013 to 2020 (a); averaged monthly mean UV radiation in mainland China in 2005–2020 (b); average seasonal mean UV radiation in mainland China in 2005–2020 in spring (c), summer (d), autumn (e), and winter (f).

**4 Discussion**

This study developed a random forest model using a variety of predictors to predict daily UV radiation in mainland China with

relatively high accuracy, resolution, and spatiotemporal coverage. Temporal and spatial characteristics were identified based on the predictions generated from the model. A gradual increase in UV radiation in recent years was observed, with an uneven spatial distribution.

This study predicted UV radiation based on a machine learning algorithm at a daily level and 10 km spatial resolution with nearly full coverage in China using multiple predictors, including satellite and simulated UV radiation data. The $R^2$ (RMSE) between measured and predicted UV radiation was 0.97 (15.64 W m$^{-2}$) for model development and 0.83 (37.44 W m$^{-2}$) for overall 10-fold CV at a daily level. Compared to other environmental factors affecting population health, such as air pollution, few studies have developed models for UV radiation and most have been conducted in the United States and Europe using statistical models such as regression analysis and area-to-point residual kriging (Feister et al., 2008; Junk et al., 2007; Pei and He, 2019; Vopham et al., 2016). In recent years, several studies have employed machine learning algorithms such as deep neural networks, support vector machine, and tree methods to predict UV radiation (Wu et al., 2022; Zhao and He, 2022). In previous studies, $R^2$ between measured and predicted UV radiation for model development ranged from 0.92 to 0.98 (Liu et al., 2017; Zhao and He, 2022; Qin et al., 2020), which were comparable with our results. In this study we employed random forest method to develop the models as it is a widely used machine learning algorithm with several advantages for predicting multiple environmental factors (Araki et al., 2018; Guo et al., 2021; Huang et al., 2018; Liu et al., 2020). First, random forest exhibits high flexibility in processing various types of data and strong tolerance to multicollinearity among predictors (Breiman, 2001; Fox et al., 2017; Strobl et al., 2008; Bamrah et al., 2020). Second, comparing to some other black-box machine learning models, random forest method is able to provide feature importance rankings and facilitate a deeper understanding in contribution of all predictors in predictions, which makes the models easier to be understood and explained (Hu et al., 2017; Wei et al., 2019). Third, the predicting errors in random forest models are generally lower, due to the reduction in variance achieved by aggregating multiple trees (Ameer et al., 2019; Ding and Qie, 2022). Forth, random forest is user-friendly with relatively small number of parameter settings and a relatively fast processing speed (Ameer et al., 2019; Hu et al., 2017). Due to the above advantages, many previous studies found that random forest method could achieve higher or at least comparable predicting accuracy over other machine learning models in predicting environmental factors (Liang et al., 2020; Julián et al., 2015; Contreras and Ferri, 2016; Ameer et al., 2019). In this study, we also compared results from random forest model and eXtreme Gradient Boosting (XGBoost) model, which is another machine learning model based on decision trees with relatively high predicting accuracy (Zamani Joharestani et al., 2019; Nasabpour Molaei et al., 2023; Dai et al., 2023; Wu et al., 2022). The results indicated that the predicting accuracy from XGBoost method was comparable but slightly lower than those from random forest method with lower $R^2$ (XGBoost: 0.81 v.s. random forest: 0.83) and higher RMSE (XGBoost: 39.25 W m$^{-2}$ v.s. random forest: 37.44 W m$^{-2}$). Several studies have developed models to predict UV radiation in China; however, the role of satellite UV radiation measurements in model performance has not been investigated. UV radiation data from satellites have

proven to be an effective variable for evaluating exposure levels and identifying hotspots of skin cancer risk in other countries (Zhou et al., 2019; Kennedy et al., 2021). Satellite-sourced UV radiation data, such as OMI EDD, offer a form of direct measurements of UV radiation from satellites, providing "real values" to constrain UV radiation predictions during spatial extrapolation (Gholamnia et al., 2021). Including the OMI EDD in the UV radiation model improved the prediction accuracy by approximately 2% compared to the model without it in this study. Additionally, this study filled in the missing values of OMI EDD data to make the spatiotemporal coverage of UV radiation predictions close to 100%, which was higher than previous studies that predicted UV radiation at 724 conventional meteorological stations in China or those that did not address the missing values in UV radiation predictions caused by incomplete predictor variables, such as AOD data from remote sensing (Wu et al., 2022; Liu et al., 2017). Gridded UV radiation predictions with nearly full spatiotemporal coverage can provide more comprehensive and flexible support for exposure assessment in health studies on exposure windows and geographic locations.

The results indicated that UV radiation is unevenly distributed throughout China, with high-exposure areas primarily located in the southwest and health-risk hotspots primarily located in the eastern region. The spatial distribution of UV radiation is closely correlated with elevation, latitude, and climatic factors. Higher elevations result in stronger UV radiation, primarily because of the thinner atmosphere, meaning that less UV radiation is absorbed or scattered by the atmosphere (Blumthaler et al., 1997). The UV radiation intensity also increases with decreasing latitude, primarily because regions at low latitudes have a smaller SZA (Holzle and Honigsmann, 2005). The spatial distribution of UV radiation in autumn effectively reflects its correlation with elevation and latitude. Meteorological factors affect UV radiation intensity. For example, cloud cover can absorb and scatter UV radiation (Dieste-Velasco et al., 2023). The higher cloud cover and humidity in subregion D resulted in higher UV radiation at F than at D, despite their similar elevations and latitudes (Fig. 5). In spring, due to factors such as air currents, the southern regions are subjected to increased precipitation, which results in elevated cloud cover and humidity (Yao et al., 2017). Consequently, this phenomenon may have resulted in lower UV radiation intensity in the southern regions than in the relatively arid northern regions. In addition to natural factors, population distribution should be considered when identifying health-risk hotspots. Although UV radiation levels were medium-high in the southeastern coastal regions, the population health effects due to UV radiation should not be ignored because of the high population density there. The threshold for the health effects of UV radiation on the population is still unclear, and there are no atmospheric UV radiation standards so far, which requires support from further epidemiological studies. The UV radiation predictions in this study covered the entire geographical area of mainland China, providing exposure data to support health studies in different regions and further identify the health risk hotspots of UV radiation exposure in China.

The UV radiation levels exhibited both seasonal and long-term temporal trends. The seasonal pattern showed the strongest UV radiation in summer and the lowest in winter. This observed pattern may be linked to variations in daylight hours and alterations

in the SZA throughout the year (Liu et al., 2017). Specifically, our findings demonstrated an increasing trend in UV radiation since 2015, accompanied by a decrease in $PM_{2.5}$ and increase in $O_3$, suggesting a potential correlation between UV radiation levels and air pollution. The decrease in $PM_{2.5}$ may contribute to the increase in UV radiation, as $PM_{2.5}$ can absorb and reflect UV radiation (Madronich et al., 2023; Gao et al., 2013). UV radiation plays a crucial role in the production of surface $O_3$ because ground-level $O_3$ primarily originates from photochemical reactions (Guicherit and Roemer, 2000). Additionally, the results of the SHAP analysis were consistent with the long-term trend analysis, which indicated that ambient aerosols levels were negatively associated with UV radiation predictions while $O_3$ concentrations positively related with UV radiation levels. The Chinese government launched and implemented a series of nationwide policies to decrease air pollution levels, including the Action Plan of Air Pollution Prevention and Control in 2013 and Three-Year (2018–2020) Action Plan for Cleaner Air in 2017. Owing to these policies, the concentrations of several air pollutants, especially $PM_{2.5}$ have decreased significantly in China since 2013. Therefore, along with a decrease in $PM_{2.5}$, there is a need to enhance public awareness of UV radiation protection.

The relatively small number of UV radiation monitoring stations employed for model development across the national landscape may have influenced the extrapolation performance of the model. The UV monitoring stations were distributed in different geographic locations with multiple land-cover types, which helped validate the model performance in spatial extrapolation. However, a spatial CV was conducted, which only slightly decreased compared to the overall CV, showing a relatively higher accuracy of spatial extrapolation.

**5 Data availability**

The UV radiation gridded dataset across mainland China in 2005–2020 is currently freely available at https://doi.org/10.5281/zenodo.10884591 (Jiang et al., 2024).

**6 Conclusion**

This study established a machine learning model for predicting daily UV radiation levels at a 10 × 10 km spatial resolution across mainland China for 16 years. The model with satellite-sourced UV radiation measurements had a higher prediction accuracy than the one without such a predictor. Based on high-resolution and coverage predictions, a gradual increase in UV radiation in recent years and an uneven spatial distribution were observed. This study provides a modeling method and exposure data for UV radiation to support exposure assessment for future epidemiological studies and the identification of exposure risk and health risk hotspots of UV radiation in the Chinese population.

**Appendix A: Additional figures and tables**

Table A1 Statistical descriptions of UV radiation measurements from ground monitoring stations in CERN in China from 2005–2020

| Year | Mean ( W m$^{-2}$ ) | Standard deviation ( W m$^{-2}$ ) | P25 ( W m$^{-2}$ ) | Median ( W m$^{-2}$ ) | P75 ( W m$^{-2}$ ) |
|------|------|------|------|------|------|
| 2005 | 160.62 | 81.07 | 94.35 | 153.57 | 160.62 |
| 2006 | 158.34 | 80.56 | 94.20 | 149.90 | 214.90 |
| 2007 | 159.54 | 82.99 | 91.81 | 150.41 | 220.21 |
| 2008 | 162.39 | 83.09 | 93.49 | 153.60 | 223.16 |
| 2009 | 159.64 | 82.65 | 91.46 | 152.20 | 222.60 |
| 2010 | 155.46 | 81.73 | 88.56 | 144.91 | 215.80 |
| 2011 | 160.95 | 84.37 | 90.11 | 152.60 | 223.50 |
| 2012 | 159.65 | 85.38 | 88.75 | 153.60 | 221.80 |
| 2013 | 160.21 | 82.87 | 92.00 | 149.93 | 221.50 |
| 2014 | 160.87 | 82.41 | 94.06 | 152.90 | 221.50 |
| 2015 | 170.96 | 91.32 | 96.66 | 162.70 | 238.20 |
| 2016 | 175.66 | 96.84 | 97.72 | 162.75 | 248.00 |
| 2017 | 180.90 | 109.28 | 100.90 | 168.40 | 254.60 |
| 2018 | 187.00 | 103.48 | 102.00 | 176.30 | 262.00 |
| 2019 | 189.80 | 104.63 | 103.90 | 178.60 | 265.70 |

| 2020 | 190.10 | 105.01 | 104.10 | 177.20 | 266.90 |
| 2005–2020 | 168.40 | 91.39 | 94.80 | 158.10 | 232.80 |

Table A2. Missing rate of erythemal daily dose (EDD) retrieved from the Ozone Monitoring Instrument (OMI) before and after

gap-filling and the results of 10-fold cross-validation of the three-day moving average method from 2005 to 2020 in China.

| year | Missing rate before gap-filling | Missing rate after gap-filling | $R^2$ of 10-fold cross-validation |
|---|---|---|---|
| 2005 | 3.03% | 0.00% | 0.90 |
| 2006 | 3.53% | 0.27% | 0.90 |
| 2007 | 3.38% | 0.00% | 0.90 |
| 2008 | 5.69% | 0.57% | 0.89 |
| 2009 | 20.33% | 0.21% | 0.88 |
| 2010 | 30.28% | 0.40% | 0.88 |
| 2011 | 33.59% | 0.53% | 0.88 |
| 2012 | 21.80% | 0.17% | 0.90 |
| 2013 | 24.24% | 0.28% | 0.88 |
| 2014 | 28.20% | 0.37% | 0.90 |
| 2015 | 31.95% | 0.50% | 0.88 |
| 2016 | 35.29% | 4.19% | 0.87 |
| 2017 | 32.78% | 1.52% | 0.86 |
| 2018 | 32.19% | 0.55% | 0.85 |

| 2019 | 32.12% | 0.42% | 0.85 |
| 2020 | 30.34% | 0.00% | 0.86 |
| 2005-2020 | 23.04% | 0.62% | 0.88 |

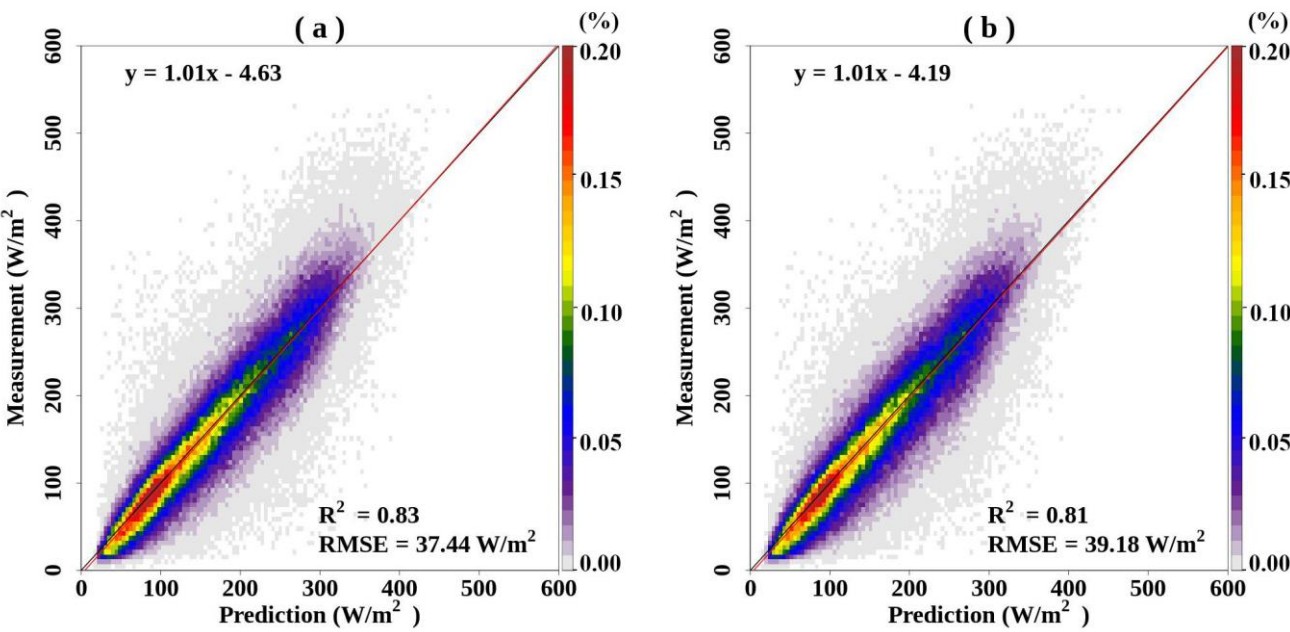

Figure A1. Density scatter plots and linear regressions between measurements and predictions of UV radiation at a daily level based on a random forest model during 2005–2020. With erythemally daily dose retrieved from the Ozone Monitoring Instrument (a) and without erythemally daily dose retrieved from the Ozone Monitoring Instrument (b).

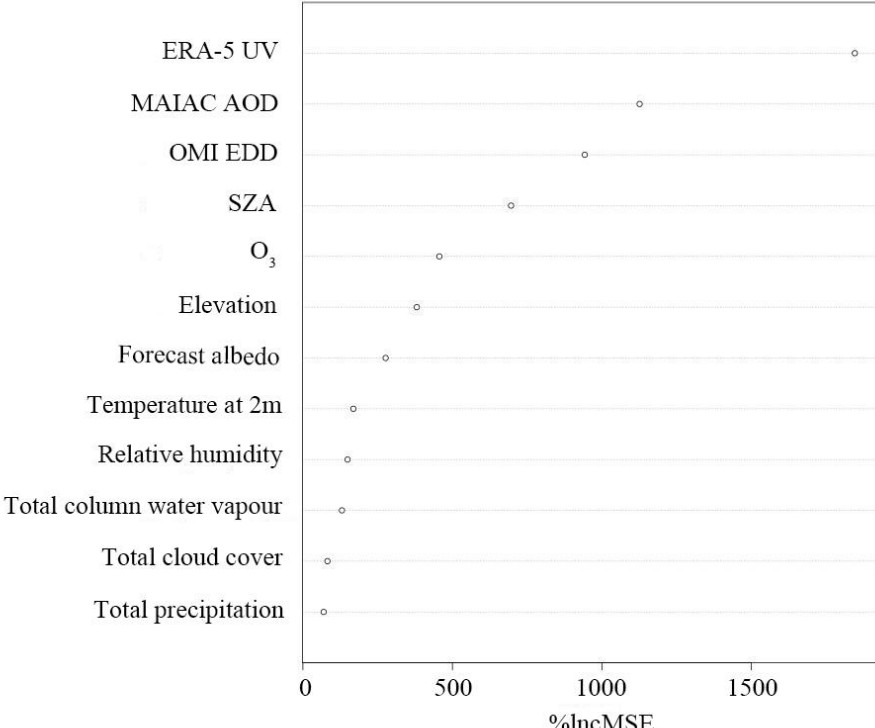

Figure A2. Ranking of importance for predictor variables in UV radiation prediction model. Note: downward UV radiation at the surface from the fifth generation European Center for Medium-Range Weather Forecasts Reanalysis (ERA-5 UV), aerosol optical depth data from the Multi-Angle Implementation of Atmospheric Correction (MAIAC AOD), erythemally daily dose retrieved from the Ozone Monitoring Instrument (OMI EDD), solar zenith angle (SZA).

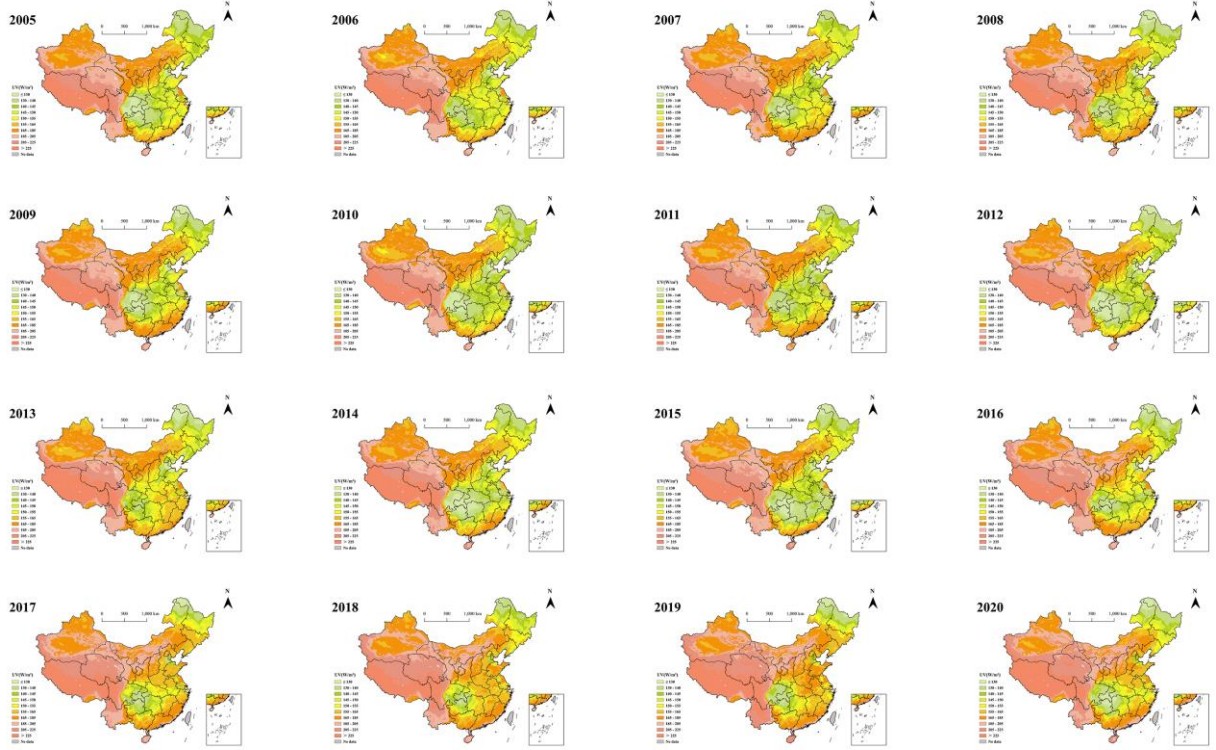

Figure A3. Spatial distributions of UV radiation based on predictions at an annual level from 2005 to 2020.

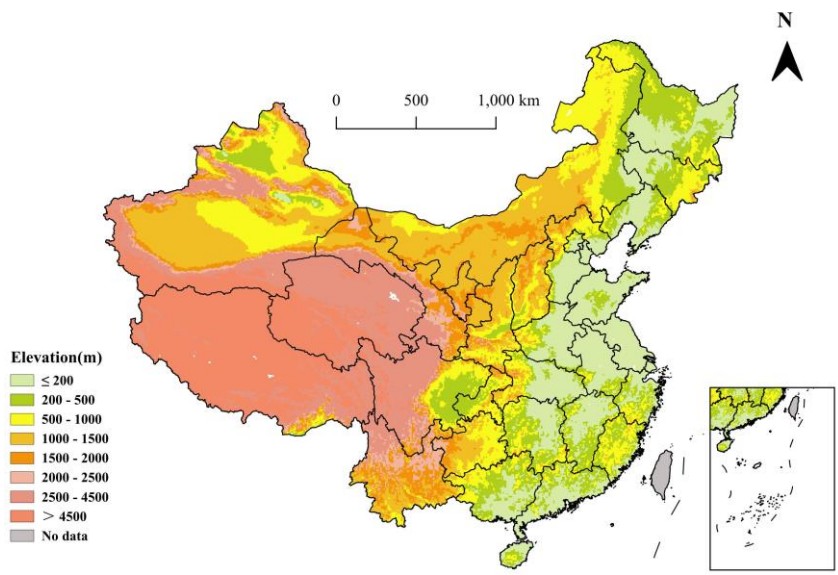

Figure A4. Spatial distribution of elevation in mainland China.

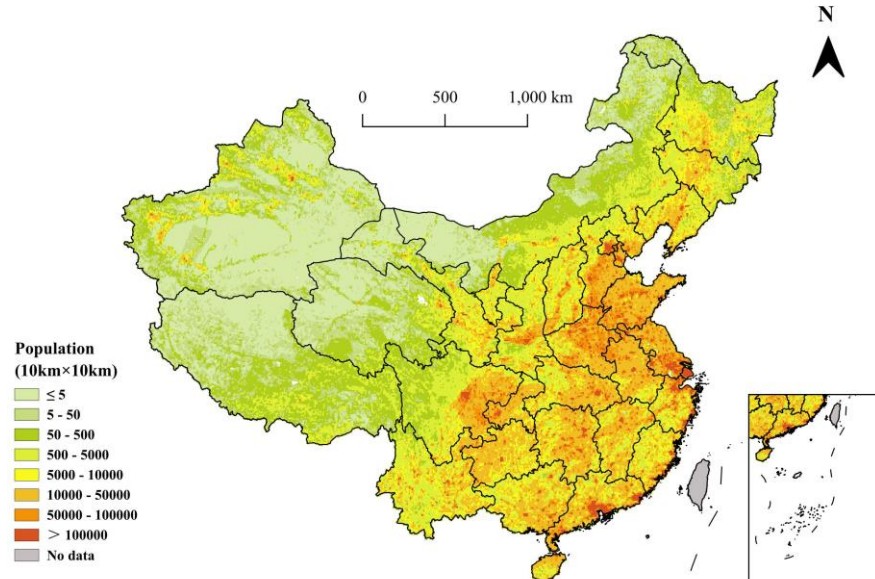

Figure A5. Spatial distribution of population in mainland China in 2020

## Author contributions

Yichen Jiang: Conceptualization, Data curation, Methodology, Software, Writing- Original draft preparation, Writing – review & editing.    Su Shi: Data curation, Software, Validation.    Xinyue Li: Data curation, Software.    Chang Xu: Data curation, Software.    Haidong Kan: Funding acquisition, Writing – review & editing.    Bo Hu: Resources, Funding acquisition, Writing – review & editing.    Xia Meng: Conceptualization, Resources, Funding acquisition, Supervision, Writing – review & editing.

## Competing interests

We declare that we have no conflict of interest.

## Acknowledgements

This work was supported by the National Key Research and Development Program of China (No. 2023YFC3708304, 2022YFC3700705); National Natural Science Foundation of China (82030103).

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
