# Peer review of "A 10 km daily-level ultraviolet radiation predicting dataset based on machine learning models in China from 2005 to 2020"

_Earth System Science Data, 2024_

## Author Comment (AC1)

**Response to Reviewer #1:**

This is a good paper which provides valuable UV dataset for epidemiological studies. In this study, they performed very strict validations using spatial, temporal, as well as by-year cross validation methods, indicating the high accuracy of their reconstructed UV dataset. I believe this dataset is valuable for environmental health studies of UV in China. I have some comments for the authors to improve the manuscript.

Response: Thank you for the positive comments and constructive suggestions to help improve our manuscript. We have fully responded to the comments below point-to-point and revised the manuscript accordingly. The line numbers referred to in this response document corresponded to those in the revised manuscript with tracked changes.

1. It is not clear why missing values of OMI EDD data have been greatly increased since 2008. Please explain it.

Response: Thank you for the suggestion. We added the explanation in lines 48-51 in the revision as "However, missing values of the OMI EDD data were non-random. Especially since 2008, the field of view of the instrument has been partially obstructed by the peeling of the spacecraft's protective film, leading to data loss in the center-right section of each observational swath. This has greatly increased the missing rate of OMI EDD data, posing a challenge to the accuracy of exposure assessments in epidemiological studies (Mcpeters et al., 2015)".

References:

McPeters, R. D., Frith, S., and Labow, G. J.: OMI total column ozone: extending the long-term data record, Atmospheric Measurement Techniques, 8, 4845-4850, https://doi.org/10.5194/amt-8-4845-2015, 2015.

2. The method using to fill the missing OMI EDD values is not clear. Specifically, what is the three-day moving average method? Does this method have enough accuracy to fill the missing values? If there are many consecutive days with missing values, how to address this?

Response: Thanks for the suggestions.

First, we added explanation of the three-day moving average method as suggested in lines 144-148 in the revision as "We employed the three-day moving average method to fill the OMI EDD values on grid-days with missing data by calculating the mean of the OMI EDD values from

the two preceding days if they were available for those grid cells. In the case of grid cells with missing data on consecutive days (more than 1 day), the missing OMI EDD data were not filled in this study. With this method, the missing rate of OMI EDD significantly decreased from 23.04% to 0.62% on average in 2005-2020 (Table A2)".

Second, we utilized 10-fold cross-validation (CV) to assess the accuracy of the three-day moving average method and added relevant descriptions and results in the revision in lines 148-153 as "10-fold CV was employed to assess the accuracy of the three-day moving average method for filling the gap of OMI EDD data. In each iteration, 10% of the original OMI EDD data in the dataset were randomly dropped, and the three-day moving average method was applied to fill the missing values. This process was repeated ten times, and the gap-filled OMI EDD values were compared to the corresponding original OMI EDD values. The results of the 10-fold CV are presented in Table A2 in Appendix, with $R^2$ ranging from 0.85 to 0.90 in 2005-2020, indicating the relatively high accuracy of the gap-filling method".

Table A2 in Appendix was modified accordingly and was displayed here for your convenient reference.

Table A2. Missing rate of erythemal daily dose (EDD) retrieved from the Ozone Monitoring Instrument (OMI) before and after gap-filling and the results of 10-fold cross-validation of the three-day moving average method from 2005 to 2020 in China.

| Year | Missing rate before gap-filling | Missing rate after gap-filling | $R^2$ of 10-fold cross-validation |
|---|---|---|---|
| 2005 | 3.03% | 0.00% | 0.90 |
| 2006 | 3.53% | 0.27% | 0.90 |
| 2007 | 3.38% | 0.00% | 0.90 |
| 2008 | 5.69% | 0.57% | 0.89 |

| | | | |
|---|---|---|---|
| 2009 | 20.33% | 0.21% | 0.88 |
| 2010 | 30.28% | 0.40% | 0.88 |
| 2011 | 33.59% | 0.53% | 0.88 |
| 2012 | 21.80% | 0.17% | 0.90 |
| 2013 | 24.24% | 0.28% | 0.88 |
| 2014 | 28.20% | 0.37% | 0.90 |
| 2015 | 31.95% | 0.50% | 0.88 |
| 2016 | 35.29% | 4.19% | 0.87 |
| 2017 | 32.78% | 1.52% | 0.86 |
| 2018 | 32.19% | 0.55% | 0.85 |
| 2019 | 32.12% | 0.42% | 0.85 |
| 2020 | 30.34% | 0.00% | 0.86 |
| 2005-2020 | 23.04% | 0.62% | 0.88 |

3. For the method of comparing the long-term trend of UV radiation and air pollution, they should use an independent section. They should not include it in the section of 2.1.4 Other predictor variables.

Response: Thanks for the suggestion. An independent section was added in the " 2.1 Data " Section:

[lines 125-129] "2.1.5 Air pollution data

For comparing the long-term trends of UV radiation and air pollution, fine particulate matter

($PM_{2.5}$) and $O_3$ data were included. $PM_{2.5}$ data were predicted using a random forest model at a daily level and a spatial resolution of 1 × 1 km in China (Meng et al., 2021; Shi et al., 2023a; Shi et al., 2023b). The source and spatiotemporal resolution of the $O_3$ data were the same as those in Section 2.1.4 Other predictor variables".

4. More analyses about the relationship between PM2.5/O3 and UV should be conducted. Although they show the importance for predictor variables, which shows AOD and O3 are important variables. They should perform SHAP analysis to show the impact directions of AOD/O3 on the UV. This could further demonstrate the impacts of PM2.5/O3 on UV increase. Response: Thanks for the constructive comment. We have conducted SHAP analysis to further elucidate the impact direction of predictors on UV radiation predictions as suggested, and added relevant descriptions in Method and Results sections, which were also summarized below for your convenient reference.

Descriptions of the methods were added in the " 2.2 Methods " Section in lines 178-188 as:

"2.2.3 Impacts of predictors on UV predictions

Two methods were applied to evaluate the impacts of all predictors on UV radiation levels. First, random forest model itself could produce importance rankings of all predictors to evaluate the contribution of each predictor to UV radiation predictions, and this is also one of the advantages of the random forest model. The importance of a predictor was measured by randomly permuting its values and comparing the decrease in predicting accuracy between the predictions before and after the permutation. Second, SHapley Additive exPlanations (SHAP) method can be used to evaluate both contributions and directions of predictors on final predictions (Lundberg and Lee, 2017). SHAP method employs the classic game theory concept of Shapley values to compute the feature importance for a specific machine learning model (Strumbelj and Kononenko, 2010). Aggregating the SHAP values across multiple data points provides a global explanation of the model. In this study, we utilized the SHAP library in Python to interpret impacts of predictors on UV radiation predictions based on a random forest model (Lundberg

et al., 2020).”

Relevant results of SHAP method were added in lines 222-233 as:

“3.3. Impacts of predictors on UV radiation predictions

Fig. A2 shows the importance ranking of all predictors produced by the random forest model itself that ERA-5 UV, OMI EDD, and MAIAC AOD were the most important predictors of UV radiation. Fig. 4 shows the SHAP summary plot and feature importance, which were the same with that from the random forest method. SHAP method also provided evaluation on the impact directions of predictors on UV radiation predictions. In Fig. 4a, each point represents a sample from the dataset. The color of each point indicates the magnitude of the predictor, with redder values indicating higher values and bluer indicating lower values. For example, ERA-5 UV and OMI EDD exerted the most substantial impact and similar impact directions on UV radiation predictions. High values of ERA-5 UV and OMI EDD increased the predicted UV radiation predictions, whereas low values decreased UV radiation predictions. Ambient aerosols (MAIAC AOD) and $O_3$ levels showed opposite effects on UV radiation predictions based on SHAP method. Higher MAIAC AOD values displayed higher negative SHAP values, meaning that higher MAIAC AOD values tended to associate with decreased UV radiation levels. Conversely, High $O_3$ levels corresponded to positive SHAP values, indicating that high $O_3$ levels were associated with high UV radiation predictions.”

[Figure]

Figure 4. Impacts of predictors on UV radiation predictions based on SHAP method (a); importance ranking of predictors for predicting UV radiation levels, calculated by taking the average of the absolute SHAP values (b).

Relevant discussions were added in " 4 Discussion " Section in lines 343-345 as:

"Additionally, the results of the SHAP analysis were consistent with the long-term trend analysis, which indicated that ambient aerosols levels were negatively associated with UV radiation predictions while $O_3$ concentrations positively related with UV radiation levels."

5. Table 1 is not necessary in main text. I recommend combine Table 1 into Table A1.
Response: Thanks for the suggestion. We have combined Table 1 into Table A1 in the Appendix. We also displayed Table A1 here for your convenient reference.

Table A1 Statistical descriptions of UV radiation measurements from ground monitoring stations in CERN in China from 2005–2020

| Year | Mean ( W m$^{-2}$ ) | Standard deviation ( W m$^{-2}$ ) | P25 ( W m$^{-2}$ ) | Median ( W m$^{-2}$ ) | P75 ( W m$^{-2}$ ) |
|---|---|---|---|---|---|
| 2005 | 160.62 | 81.07 | 94.35 | 153.57 | 160.62 |
| 2006 | 158.34 | 80.56 | 94.20 | 149.90 | 214.90 |
| 2007 | 159.54 | 82.99 | 91.81 | 150.41 | 220.21 |
| 2008 | 162.39 | 83.09 | 93.49 | 153.60 | 223.16 |
| 2009 | 159.64 | 82.65 | 91.46 | 152.20 | 222.60 |
| 2010 | 155.46 | 81.73 | 88.56 | 144.91 | 215.80 |
| 2011 | 160.95 | 84.37 | 90.11 | 152.60 | 223.50 |

| | | | | | |
|---|---|---|---|---|---|
| 2012 | 159.65 | 85.38 | 88.75 | 153.60 | 221.80 |
| 2013 | 160.21 | 82.87 | 92.00 | 149.93 | 221.50 |
| 2014 | 160.87 | 82.41 | 94.06 | 152.90 | 221.50 |
| 2015 | 170.96 | 91.32 | 96.66 | 162.70 | 238.20 |
| 2016 | 175.66 | 96.84 | 97.72 | 162.75 | 248.00 |
| 2017 | 180.90 | 109.28 | 100.90 | 168.40 | 254.60 |
| 2018 | 187.00 | 103.48 | 102.00 | 176.30 | 262.00 |
| 2019 | 189.80 | 104.63 | 103.90 | 178.60 | 265.70 |
| 2020 | 190.10 | 105.01 | 104.10 | 177.20 | 266.90 |
| 2005–2020 | 168.40 | 91.39 | 94.80 | 158.10 | 232.80 |

---

## Author Comment (AC2)

**Response to Reviewer #2:**

The study developed a machine learning model to predict UV radiation and highlighted the model performance. This research topic is very important given the rise in the UV radiation in recent years.

Response: Thank you for the positive comments and constructive suggestions to help improve our manuscript. We have fully responded to the comments below point-to-point and revised the manuscript accordingly. The line numbers referred to in this response document corresponded to those in the revised manuscript with tracked changes.

Overall comments: Technically the manuscript seems strong however the writing can be improved.

Highly suggest the authors to go through the language and make changes wherever necessary throughout the manuscript.

Response: Thanks for your suggestion. We carefully reviewed and revised the language throughout the entire manuscript. Also, we have had the manuscript language-edited by Elsevier. The certificate is provided below.

[Figure]

**Certificate of Elsevier**
**Language Editing Services**

**The following article was edited by Elsevier Language Editing Services:**

A 10 km daily-level ultraviolet radiation predicting dataset
based on machine learning models in China from
2005 to 2020

**Ordered by:**

Yichen Jiang

**Estimated Delivery date:**
2024-07-30
**Order reference:**
ASLESTD1069282

[Figure]

Comments:

Line 14: Seems grammatically incorrect. Reword it to "but limited studies have implemented it for UV radiation"

Response: Thank you for the suggestion. The sentence has been revised as "Machine learning algorithms have been widely used to predict environmental factors with high accuracy, but limited studies have implemented it for UV radiation." in lines 13-14 in the revision as suggested.

Line 14-15: The language can be improved. Reword these lines to "The main aim of this study is to develop UV radiation prediction model using the random forest approach and predict the UV radiation at daily and 10km resolution in mainland China from 2005 to 2020".

Response: Thank you for the suggestion. The sentence has been revised as "The main aim of this study is to develop UV radiation prediction model using the random forest approach and predict the UV radiation at daily and 10 km resolution in mainland China from 2005 to 2020." in lines 14-16 in the revision as suggested.

Line 16: It is already mentioned above that random forest model was employed to predict UV radiation. Reword this line.

Response: Thank you for the suggestion. The sentence has been revised as "The model was developed with multiple predictors such as UV radiation data from satellites as independent variables and ground UV radiation measurements from monitoring stations as the dependent variable." in lines 16-17 in the revision as suggested.

Line 21: OMI EDD is used for the first time, write the full form of EDD before introducing the acronym

Response: Thank you for the suggestion. The sentence has been revised as "The model that incorporated erythemal daily dose (EDD) retrieved from the Ozone Monitoring Instrument (OMI) had a higher prediction accuracy than that without it." in lines 21-22 in the revision as suggested.

Line 26: Consider rewording this line as it is not flowing well. May be change it to something like this: "Using machine learning this study generated gridded UV radiation dataset with extensive spatiotemporal coverage which can be utilized for future health-related research".

Response: Thank you for the suggestion. The sentence has been revised as "Using machine learning algorithm, this study generated gridded UV radiation dataset with extensive spatiotemporal coverage, which can be utilized for future health-related research." in lines 27-28 in the revision as suggested.

Line 35 - 36: Please consider rewording these lines.

Response: Thank you for the suggestion. The sentence has been revised as "Further studies are required to ascertain the effects of UV radiation on human health; however, the lack of high-accurate exposure data of UV radiation hinders such health-related investigations." in lines 35-37 in the revision as suggested.

Line 43: remove stands, change it to "despite being"

Response: Thank you for the suggestion. The sentence has been revised as "For example, erythemal UV irradiance from the Total Ozone Mapping Spectrometer (TOMS), despite being one of the initial instruments for evaluating the UV radiation backscattered by the Earth's atmospheric layers, it exhibits lower spatial resolution of 50 km×50 km, and it has limited accuracy." in lines 42-44 in the revision as suggested.

Line 70: Remove "What's more, missingness of satellite-based", change it to" The missing satellite-based"

Response: Thank you for the suggestion. The sentence has been revised as "The missing satellite-based UV radiation were filled to improve the spatial coverage of the final UV radiation predictions." in lines 71-72 in the revision as suggested.

Line 108: Why did the author use O3 concentrations predicted from random forest and not use

directly the monitoring data? Clarify this and explain it in the text clearly.

Response: Thank you for the comments and suggestions. As suggested, we further clarified this issue in the revision in lines 110-118 as "This study used gridded $O_3$ data instead of $O_3$ monitoring data from station sites, primarily due to considerations of data coverage in both temporal and spatial dimensions. Regarding the temporal coverage, the air quality monitoring network in China has not established until 2013, which could not fully cover the study period of 2005-2020 in this study. For the spatial coverage, the density of air quality monitoring stations is relatively low, with the majority of them are located in urban areas and eastern China, which could not capture the spatial variability within city and reflect the $O_3$ pollution level in rural areas and western regions (Geyh et al., 2000). While the gridded $O_3$ predictions used in this study are available from 2005-2020, have full spatial coverage in mainland China and achieved relatively high accuracy comparing with ground measurements with cross-validation (CV) $R^2$ and root mean square error of 0.80 and 20.93 ug/m$^3$, respectively (Meng et al., 2022)."

Line 139: provide reference for 10-fold cross validation if it was used in previous studies and explain the cross-validation process details and the differences between the various (temporal, spatial and year) 10-fold CV?

Response: Thanks for the suggestion. We have added references, refined the details of the CV process, and explained the differences among various CVs in lines 155-177 in the revision as "
[revised manuscript text omitted]

Line 123: Why did the authors use random forest compared to the other machine learning algorithm? Include the necessary information that supports the argument.

Response: Thanks for the constructive comment.

Overall, random forest method is a widely used machine learning algorithm for predicting multiple environmental factors (Araki et al., 2018; Guo et al., 2021; Huang et al., 2018; Liu et al., 2020), with several advantages comparing with other machine learning methods. First, random forest exhibits high flexibility in processing various types of data and strong tolerance to multicollinearity among predictors (Breiman, 2001; Fox et al., 2017; Strobl et al., 2008; Bamrah et al., 2020). Second, comparing to some other black-box machine learning models, random forest method is able to provide feature importance rankings and facilitate a deeper understanding in contribution of all predictors in predictions, which makes the models easier to be understood and explained (Hu et al., 2017; Wei et al., 2019). Third, the predicting errors in random forest models are generally lower, due to the reduction in variance achieved by aggregating multiple trees (Ameer et al., 2019; Ding and Qie, 2022). Forth, random forest is user-friendly with relatively small number of parameter settings and a relatively fast processing speed (Ameer et al., 2019; Hu et al., 2017). Due to the above advantages, many previous studies found that random forest method could achieve the higher or at least comparable predicting accuracy over other machine learning models. A study in Taiwan, China, predicting the air quality index showed that compared to methods of adaptive boosting, artificial neural networks, stacking ensemble, and support vector machines, the random forest model performed better (Liang et al., 2020). In a study predicting CO, NO, $PM_{2.5}$, and $NO_2$ in Spain, random forest outperformed other machine learning models (decision tree for regression, support vector machines, and neural networks) in predicting almost all pollutants (Ochando et al., 2015). A study compared multiple models of decision tree, random forest, gradient boosting, and artificial neural network multi-layer perceptron in predicting $PM_{2.5}$ in multiple cities in China

and found that random forest model performed the best (Ameer et al., 2019). In another study conducted in Valencia, Spain, comparing a decision tree for regression and random forest in predicting NO, $NO_2$, $SO_2$, and $O_3$, the random forest model produced better results (Contreras and Ferri, 2016). A study in India predicting the Air Quality Index compared decision tree, support vector regression, and random forest, with random forest having the highest accuracy (Bamrah et al., 2020).

In the revision, we also added an extra analysis by developing an eXtreme Gradient Boosting (XGBoost) model, another machine learning model based on decision trees with relatively high predicting accuracy (Zamani Joharestani et al., 2019; Nasabpour Molaei et al., 2023; Dai et al., 2023; Wu et al., 2022). Based on data in this study, the XGBoost model yielded an $R^2$ (RMSE) of 0.81 (39.25 W $m^{-2}$) in predicting UV radiation levels with the same predictors in the random forest model, while the random forest model achieved better performance with slightly higher $R^2$ (0.83) and lower RMSE (37.44 W $m^{-2}$). Therefore, this study employs random forest to construct the models.

[revised manuscript text omitted]

Line 218: Fix the typo. It is Figure 5 not 3.

Response: Thanks for pointing out the issue. It has been corrected.

Line 269: reword the line to "there is no atmospheric UV standards"

Response: Thank you for the suggestion. The sentence has been revised as "The threshold for the health effects of UV radiation on the population is still unclear, and there are no atmospheric UV radiation standards so far, which requires support from further epidemiological studies." in lines 332-334 in the revision as suggested.